# The Role of Drones in Out-of-Hospital Cardiac Arrest: A Scoping Review

**DOI:** 10.3390/jcm11195744

**Published:** 2022-09-28

**Authors:** Joseph Chun Liang Lim, Nicole Loh, Hsin Hui Lam, Jin Wee Lee, Nan Liu, Jun Wei Yeo, Andrew Fu Wah Ho

**Affiliations:** 1Yong Loo Lin School of Medicine, National University of Singapore, Singapore 117597, Singapore; 2Centre for Qualitative Medicine and Programme in Health Services and Systems Research, Duke-NUS, Singapore 169857, Singapore; 3SingHealth AI Health Programme, Singapore Health Services, Singapore 168753, Singapore; 4Institute of Data Science, National University of Singapore, Singapore 119077, Singapore; 5Department of Emergency Medicine, Singapore General Hospital, Singapore 168753, Singapore; 6Pre-Hospital and Emergency Research Centre, Duke-NUS Medical School, Singapore 169857, Singapore

**Keywords:** automated external defibrillators, emergency medical services, out-of-hospital cardiac arrest, unmanned aerial devices

## Abstract

Drones may be able to deliver automated external defibrillators (AEDs) directly to bystanders of out-of-hospital cardiac arrest (OHCA) events, improving survival outcomes by facilitating early defibrillation. We aimed to provide an overview of the available literature on the role and impact of drones in AED delivery in OHCA. We conducted this scoping review using the PRISMA-ScR and Arksey and O’Malley framework, and systematically searched five bibliographical databases (Medline, EMBASE, Cochrane CENTRAL, PsychInfo and Scopus) from inception until 28 February 2022. After excluding duplicate articles, title/abstract screening followed by full text review was conducted by three independent authors. Data from the included articles were abstracted and analysed, with a focus on potential time savings of drone networks in delivering AEDs in OHCA, and factors that influence its implementation. Out of the 26 included studies, 23 conducted simulations or physical trials to optimise drone network configuration and evaluate time savings from drone delivery of AEDs, compared to the current emergency medical services (EMS), along with 1 prospective trial conducted in Sweden and 2 qualitative studies. Improvements in response times varied across the studies, with greater time savings in rural areas. However, emergency call to AED attachment time was not reduced in the sole prospective study and a South Korean study that accounted for weather and topography. With growing interest in drones and their potential use in AED delivery spurring new research in the field, our included studies demonstrate the potential advantages of unmanned aerial vehicle (UAV) network implementation in controlled environments to deliver AEDs faster than current EMS. However, for these time savings to translate to reduced times to defibrillation and improvement in OHCA outcomes, careful evaluation and addressing of real-world delays, challenges, and barriers to drone use in AED delivery is required.

## 1. Introduction

### Background

Out-of-hospital cardiac arrest (OHCA) is the most time-critical medical emergency. With an incidence of 147 per 100,000 ED presentations [1] and 8.8% of these surviving to discharge [2], OHCA exerts a significant disease burden globally. Successful resuscitation can potentially avert certain death and may allow patients to return to an active life in the community [3]. Early defibrillation significantly improves survival in OHCA [4,5], as the odds of survival after bystander defibrillation raised by 2.30 times in Singapore. An unsolved challenge is how to optimise public access defibrillation (PAD) programs, as automated external defibrillators (AED) must be strategically placed to provide timely access to large populations, while remaining cost-effective [6,7]. Drones, or unmanned aerial vehicles, have emerged as a potential solution.

Early defibrillation is one of the main factors in improving survival outcomes [8]. In OHCA with shockable rhythms, the odds of survival decrease by 10% with each minute that passes without defibrillation [7,9]. PAD enables bystanders to initiate early defibrillation before the arrival of emergency medical services and improves outcomes. However, PAD programs face challenges, as the majority of OHCAs occur in private areas, such as residences [10], or at timings where the AED may not be accessible [11]. The occurrence of the coronavirus disease (COVID-19) pandemic has also resulted in a significant reduction in bystander AED use [12,13]. Combined with increased EMS response times, mortality rates of OHCA events have significantly increased [14,15,16]. As such, novel methods to optimise PAD programs are needed.

Drones are small aircrafts that can be operated remotely without human crew on board. Development in drone technology has increased its capabilities, expanding its use from the military to other areas, such as aerial surveillance, cargo transport, humanitarian relief and healthcare [17,18]. The use of UAVs to deliver life-saving medical devices such as AEDs has been a growing field of interest over the past decade. Drones may be able to deliver the AEDs directly to bystanders before local EMS arrival, thus expediting the time to defibrillation. This may also minimise time required by bystanders to perform ground search for AED. However, an assessment of the overall impact of AED-drone delivery in OHCA events and evaluation of the current literature on the role of drones in OHCA have not yet been carried out. Thus, the aim of this review is to provide a scope of the potential impacts and current state of AED drone delivery research.

## 2. Methods

This scoping review adhered to the Preferred Reporting Items for Systematic Reviews and Meta-analyses Extension for Scoping Reviews (PRISMA-ScR) and the methodological framework for scoping reviews proposed by Arksey and O’Malley [19]. Given the heterogeneity of studies that explore the use of drones in OHCA, the decision was made to synthesise the existing literature through a scoping review. This review aims to map the range of the existing literature, identify key research findings or gaps in existing knowledge, and highlight future research directions.

### 2.1. Search Strategy

We systematically searched five bibliographical databases (Medline, EMBASE, Cochrane CENTRAL, PsychInfo, and Scopus) from inception until 28 February 2022. The search strategy and choice of databases was designed in consultation with a medical information specialist (Medical Library, National University of Singapore, Singapore). To retrieve relevant articles, we used keywords and MeSH terms, including “Drones”, “Unmanned Aerial Vehicles”, “Automated External Defibrillator Delivery”, “Out-of-hospital Cardiac Arrest” and their synonyms. We also hand-searched the bibliographies of reviews that addressed related topics, such as the role of drones for health purposes, to identify further relevant articles. We also consulted subject matter experts to identify additional relevant articles. After removing duplicates, we conducted an article assessment using Endnote X9 (Clarivate, Philadelphia, PA, USA) to assess the titles and abstracts of the retrieved articles. Full texts were assessed for articles of interest. The detailed search strategy may be found in Appendix A.

### 2.2. Inclusion and Exclusion Criteria

Three authors (JCLL, HHL, NL) conducted the article assessment using predefined criteria. Each article was reviewed by at least two authors and the decision to include or exclude each article was blinded among them. Disputes were resolved through discussion and consensus with a senior author (AFWH). Articles that addressed the application of drones in OHCA were considered eligible for inclusion, where drones were defined as remotely controlled aircraft without any humans on board. Interventional trials, retrospective cohort, or prospective cohort study designs were included, as well as qualitative studies, or studies using simulation or mathematical models. We included conference abstracts to comprehensively assess the literature, referencing them as such. Studies were excluded if there was no primary datum, such as systematic reviews and meta-analyses, narrative reviews, and protocols. They were also excluded if they were non-English and without an English translation.

### 2.3. Data Abstraction

Three authors (JCLL, HHL, NL) abstracted data using a predetermined data collection form. The data abstraction process was independent and blinded among the study authors, and disputes were resolved through consensus with the senior author (AFWH). Article information (author, year of publication, country), methodology (interventional, simulation, qualitative), drone specifications (drone type, maximum range, maximum velocity), and relevant quantitative or qualitative results were abstracted. We presented continuous data in mean and standard deviation (SD) or median and interquartile range (IQR) and categorical data in percentages. Specific outcomes of interest included time saved due to AED delivery by drones compared to EMS, defined as the difference in response time between drones and EMS, the number and distribution of drone base locations according to geographical information systems (GIS) modelling, with respect to the maximum coverage location problem (MCLP), and qualitative barriers or enablers towards using drones in OHCA among bystanders and EMS personnel. Data on survival with good neurological outcome or survival alone to the time points of discharge or 30 days were also extracted whenever possible.

## 3. Results

### 3.1. Literature Retrieval and Summary of Included Articles

The literature search retrieved 970 articles. After removal of 297 duplicate articles, 626 articles were excluded based on their title and abstract. A further 24 articles were excluded based on the full text review. Finally, 26 articles were eligible for qualitative synthesis. The study selection process and reasons for exclusion is presented in the PRISMA-P 2020 Flow Diagram (Appendix A). Nine papers investigated the outcomes of different drone modelling approaches, seven papers investigated time saving of AED delivery via drones vs. current EMS and one study compared response times of AED delivery via drones vs. ground search for public AEDs. Four papers investigated the effects of meteorological conditions on drone delivery of AED, six papers examined the factors that impact the feasibility of drone delivery, and five papers examined the cost-effectiveness of drone network implementation. Finally, five papers qualitatively assessed user experience and stakeholder attitudes to implementation of AED delivery via drones.

Four studies [20,21,22,23] were conducted in Canada, one [24] in France, two [25,26] in Germany, five [27,28,29,30,31] in Sweden, two [32,33] in the United Kingdom, one [34] in Ireland, one [35] in South Korea and 10 [18,36,37,38,39,40,41,42,43,44] in the United States. These locations varied in geographic scale from counties and towns to entire cities and covered a broad range of urban and rural settings. One study [41] was a prospective trial, two studies [22,42] had a purely qualitative design that involved interviews or focus group discussions, and the rest of the twenty-three studies were simulation studies. Publication years ranged from 2016 to 2021. The characteristics of included studies may be found in Table 1.

### 3.2. Types of Drones

A total of 14 studies reported on the brand and specifications of drones used in their studies. The drones were fixed wing or multirotor systems, with maximum velocities ranging from 48.3 km/h to 100 km/h, and maximum ranges from 6 km to 80.5 km. The summary of drone models and specifications may be found in Appendix A.

### 3.3. Time Saving with AED-Delivery Using Drones

The primary application of drones in all the included studies was to deliver an AED to the site of an OHCA patient, while the most frequently measured outcome was time savings. A total of 7 studies investigated time savings of AED delivery compared to current emergency medical services. The summary of results may be found in Table 2.

Multiple simulated studies concluded that there was an overall reduction in response times when drones were used in rural areas. The extent of improvement varied widely among papers [21,24,29]. The sole prospective trial found that drones arrived prior to EMS in 64% of cases, but also that AED shock was not delivered before EMS arrival on scene in any of the cases [30].

However, response times in urban areas did not improve by a significant margin. Drones arrived before EMS services in 32% of cases (urban) vs. 93% (rural) and reported less improvement in response times [27,34].

Drone delivery was found to be favourable in the ground search for AEDs where public AEDs were not easily accessible, but ground search was favourable when public AEDs were easily accessible [41].

### 3.4. Optimal Drone Positioning

Nine studies investigated and compared the outcomes of different optimisation models to determine the placement and number of bases and drones in an implemented drone network. Use of these models to optimise drone placement at new sites improved response times compared to placement at the existing EMS sites from 80.1% to 90.3% [36].

Drone delivery of AED from regionally optimised models reduced travelling distance and time to arrival compared to EMS [21]. Integrated location-queuing optimisation models, when compared to region-specific models, required less bases and drones, and improved median time to AED [20].

Selective activation of drones via the dispatch rule allowed maintained improvements in response times, with up to 30% fewer dispatches with high accuracy [23].

Increasing the weightage of backup coverage that allowed EMS facilities to respond to a second event in its service area required a greater number of drones and reduced primary coverage [37]. A backup weight of 0.2 was found to minimise this loss of primary coverage, while increasing backup coverage significantly.

### 3.5. Feasibility and Cost-Effectiveness

Six studies assessed the practicality of use of drones for AED delivery in the real world and their barriers, which include legislation, technical issues and variability in terrain. Four studies emphasised the importance of weather and meteorological conditions in the operation of drone networks. The summary of factors that impact feasibility may be found in Table 2.

Five studies examined the expenditure required to purchase and maintain a drone network. The most cost-efficient way to achieve 90% response within 1 min was to station drones at 39 EMS sites and 12 new locations for a total of 51 sites, costing SGD 2,010,000 [36]. In addition to the initial cost, annual maintenance fees may cost 20% of the initial sum [34,38].

### 3.6. Perception of Drone Use in AED Delivery

A total of 5 studies conducted practical simulations with AED delivery via drones to participants performing CPR on a manikin, following up with qualitative interviews. Another study [22] evaluated public perception and acceptability of a drone AED delivery program ‘AED on the Fly’ in the town of Caledon, Ontario. The qualitative results from the abovementioned studies are documented in Table 3.

These studies all reported positive community attitudes to the delivery of AEDs via drones. Participants and key stakeholders alike perceived value in the potential advantages of this drone delivery system in reducing response times in OHCA events, especially in less accessible locations [26,41,42,43,44]. However, some participants reported neutral feelings, uncertainty, and anxiety towards interacting with the drone, as well as safety concerns [44].

Challenges and barriers to successful drone AED deployment were highlighted. AED and CPR usage, in addition to technology literacy limitations, were cited as significant obstacles; users reported technical difficulties in electrode attachment and placement and even in mobile phone usage [28]. Key stakeholders cited logistical, financial, legal and safety challenges, as well as concern regarding public use of AEDs [42].

Suggestions for improvement encompassed methods to facilitate bystander use, thereby decreasing the time to defibrillation. Visual and audio indicators may decrease the time required to locate the drone-delivered AED [28]. Upon delivery and location of the AED, short and clear dispatcher instructions may improve technique and compliance in AED operation [44]. Successful community engagement and implementation requires clear and consultative communication with the community during the development of the AED drone network programme [22]. Important considerations also include solidifying partnerships with relevant stakeholders, such as EMS and fire services, and identifying stable funding, as well as learning from existing drone models [42].

## 4. Discussion

This scoping review, conducted with five bibliographical databases, yielding 26 relevant articles, provided an overview of the available literature on the use of drones in AED delivery in OHCAs. The majority of studies conducted utilised software simulations to assess the time to delivery of AEDs delivered by drones against conventional EMS, using a variety of location models to determine the positioning and distribution of drones and drone bases. This scoping review is the most current representation of the varied published literature, including quantitative and qualitative studies, regarding drone delivery of AEDs in OHCA. It is an important contribution that may guide the interventional trials needed to confirm the effectiveness of drones for delivering AEDs, which remains a promising venture, as care for OHCA patients develops in the future.

All the included studies demonstrated varying, but significant, time reductions in AED delivery via drones as compared to existing EMS and may decrease time to defibrillation to increase survivability in OHCA events. Studies conducted by Bogle et al. and Pulver et al. estimate the cost of establishing a drone network at 1.3 million every 4 years and 2.01 million, respectively. The former study calculated the cost per QALY to be SGD 1937, far below the Institute for Clinical and Economic Review’s valuation of SGD 50,000–SGD 150,000 per QALY; suggesting the cost effectiveness of the implementation of a drone network. Furthermore, advancements in drone technology are promising, with more recent models of the DJI drones being used in three of the studies that showcased twice the flight time, infrared cameras with rangefinder functionality, and expanded operating temperatures of −20 to 50 degrees Celsius.

Most notably, the sole prospective trial found that in 64% of drone dispatches, the drone was able to arrive prior to EMS, with a median time difference of 1:52 min [30]. This trial provided unique insight into real-world integration of a drone network into existing EMS, accounting for the full complement of delays and challenges. While these results are a promising representation of the benefits regarding time to availability of AED, the trial noted no AEDs attached prior to ambulance arrival. Furthermore, the study found 74% of total OHCA cases ineligible for drone dispatch, due to reasons including adverse weather conditions, no-fly zones and technical difficulties.

This highlights that for the effective utilisation of drones for AED delivery in OHCA, determinants that can be broadly classified under the following subcategories must first be identified and then solved.

First, appropriate community education and literacy in AED and CPR is paramount, as is adequate knowledge and acceptance of the use of drone technology in the provision of lifesaving aid in OHCA. Currently, key barriers to public access defibrillation include the following: the majority of OHCAs occur in locations unsuitable for timely public-access AED deployment, and lack of bystander literacy and confidence in AED usage [45]. The former is the limiting factor against early bystander defibrillation in OHCA events [46,47]. Drone delivery may allow AEDs to be available at the site of OHCAs more promptly, but concurrent improvements in poor AED literacy rates [28,43] among bystanders is necessary to minimise bottlenecking of early bystander defibrillation rates by low AED usage once delivered. This may be executed through a range of methods, including lay instructors, self-directed learning and brief training. A European study found heterogeneity in AED infrastructure and legislation across different countries in Europe, which was reflected in corresponding differences in AED use and OHCA survival [48]. This highlights not only the challenges in ensuring the use of AEDs upon delivery on-scene across different locations, but that improvements in AED literacy and integration may translate to increased bystander-performed AED resuscitation.

Second, civil aviation regulations regarding drone flight must be conducive to the use of a drone network in the delivery of AEDs. Legal restrictions, including prohibition of flight out of line of sight and no-fly zones as reported by Bauer et al., are prohibitive to the development of a drone network to deliver AEDs to OHCAs. Recently, new regulations in the Aviation Law Act in Poland, for example, introduce standardised requirements for drone use, including registration of drones, weight and height limitations, anti-collision and emergency procedural technology. A challenge that remains is the requirement to obtain airspace clearance and permission before flight and its associated delays, especially in emergency use.

Third, practical considerations in operation and maintenance of a drone network include prohibitive effects of weather effects on drone flight reported in the prospective trial conducted by Schierbeck et al. Rain, specifically, prohibited drone dispatch in 8 out of 53 OHCA cases. Furthermore, technical issues, such as maintenance and battery charging, may increase the number of drones required per base for dispatch. This may be solved via a queuing model proposed by Boutilier et al., or the backup coverage location problem proposed by Pulver et al. Especially in urban contexts, high-rise buildings pose a challenge to drone delivery of AEDs as a physical obstacle to flight paths, adding complexity and delays in AED delivery to OHCAs that occur within their premises.

Other important factors to increase the effectiveness of drone-delivered AEDs were also suggested, including, but not limited to, the use of short encouraging instructions from dispatchers and increasing AED visibility via installing headlights on the drone hovering over the AED.

Finally, the most appropriate location model and cost breakdown must be determined to allow maximum utility of the drone network with the most effective coverage of the required area. This includes more recent variations in these models, including those that consider queuing, backup coverage and evaluation of whether drones should be dispatched in each scenario. This changes depending on factors such as the distribution of OHCAs, with Boutilier et al. finding that integrated networks require fewer bases and drones to achieve the same reduction in the 90th percentile of time to AED arrival as region-specific models, but may result in a loss of rural coverage.

## 5. Strengths and Limitations

This review is the most current representation of the varied published evidence related to the time savings, implementation methods, location models and challenges in the use of drones to deliver AEDs in OHCA. The strengths of this review are the systematic search technique, precise inclusion and exclusion criteria, and careful data extraction and representation process. Notably, this review included both quantitative and qualitative studies to deliver a clearer image of the state of the current literature on this novel topic.

Limitations include the inclusion of only English language literature in the review. Due to the limited number of studies performed and the heterogeneous nature of the studies in their methodologies, simulation models and outcomes, a scoping rather than systematic review was conducted. Hence, risk of bias and quality assessments of the included studies were not performed, and we were unable to draw comparisons across studies.

As drone delivery of AEDs remains a novel intervention strategy, the majority of studies conducted were simulations, with only one prospective trial conducted. There are, therefore, limited data from real-world implementation of drone networks, which is imperative in obtaining an understanding and drawing concrete conclusions regarding the multiple factors that influence the effectiveness of this solution.

## 6. Conclusions

In this scoping review of the available literature on the use of drones to facilitate AED delivery in OHCAs, the simulations and trials conducted have provided evidence that in a controlled environment, drones can deliver an AED faster than the current EMS services, thereby decreasing time to defibrillation and improving OHCA outcomes. However, drone delivery of AEDs can only be effective if supporting factors such as local aviation regulations, community education and AED literacy, for example, are evaluated together in a cohesive manner.

## Figures and Tables

**Table 1 jcm-11-05744-t001:** Characteristics of Included Studies.

Article	Country	Setting	Study Design	Total N	Dataset	Methodology
Claesson et al. (2016)	Sweden	Stockholm County, rural and downtown areas	Simulation	3165	-	GIS model used for drone base placement in rural and urban areas, comparing time taken for arrival between EMS vs. drones.
Pulver et al. (2016)	United States	Salt Lake County, Utah	Simulation	-	2010 Census Galea et al. (2002–2003)	GIS and MLCP model to determine best configuration of drones, comparing estimated travel times of EMS vs. drones at EMS locations vs. drones at new sites.
Rachunok et al. (2016)	United States	Mecklenberg County, North Carolina	Simulation	-	Mecklenberg County North Carolina	Survival probability and response time averages of EMS calculated and compared against UAV dispatch from 168 potential sites following dispatch rules.
Claesson et al. (2017)	Sweden	Norrtalje municipality, north of Stockholm, restricted airspace	Simulation	18	Swedish Registry for Cardiopulmonary Resuscitation (2006–2014)	Dispatch to locations identified for historical OHCA within 10 km of fire station, dispatch to arrival time compared between drones placed at fire stations vs. EMS.
Pulver et al. (2018)	United States	Salt Lake City, Utah	Simulation	-	Utah Department of Health Bureau of Emergency Medical Services	BLCP-CC model to identify optimal drone sites, comparing different models with different weightages for partial coverage and backup coverage of distributed demand.
Bogle et al. (2019)	United States	North Carolina, urban and rural regions across various terrains	Simulation	16,503	2009 US Census CARES	Mathematical models selected drone stations from existing infrastructure, comparing outcomes between models with 0 to 50 to 1015 docking stations.
Boutilier et al. (2019)	Canada	8 regions covered in Toronto RescuNET	Simulation	53,702	Toronto RescuNET (January 2006–December 2014)	Modelling approach to determine minimum number and location of drone bases required to improve historical median response time.
Sanfridsson et al. (2019)	Sweden	Among participants from Swedish National Pensioners’ Organisation	Practical simulation, interview	8	Swedish National Pensioners’ Organisation	Participants performed CPR on a manikin, after which an AED was delivered by drone. Qualitative and quantitative data from open interviews, observations and video recordings were analysed.
Cheskes et al. (2020)	Canada	Two rural locations in Southern Ontario (Caledon Town, Renfrew County)	Practical simulation	6	-	Call to AED attach times compared between EMS and drone dispatch from the same paramedic station vs. different paramedic station vs. optimised locations.
Glick et al. (2020)	United States	Portland, Oregon	Simulation	-	American Heart Association	Modelling framework developed to analyse drone delivery reliability by quantifying failure rates of drone AED delivery due to drone range and meteorological conditions.
Lancaster et al. (2020)	United States	Bellevue, Washington in King County; five EMS ambulance locations	Simulation	-	-	Monte Carlo sampling simulated locations of OHCAs, predicting and comparing response time of EMS vs. bystander vs. drone AED delivery. Logistic regression model used to translate response times to likelihood of survival.
Mackle et al. (2020)	Ireland	Northern Ireland	Simulation	-	HeartSine AED	Genetic algorithm determined drone base positioning, average OHCA response times calculated before and after implementation of drone network with 78 bases.
Rosamond et al. (2020)	United States	Five zones at University of North Carolina, Chapel Hill Campus	RCT, survey, interview	63	-	Participants were paired to respond to simulated OHCA with AED drone delivery. AED delivery times were compared, pre- and post-trial interviews were conducted.
Sedig et al. (2020)	Canada	Town of Caledon in Peel Region, Ontario	Interview, focus group	65	-	Purposive sampling used to recruit 40 community members. Interviews, focus group data collection and inductive thematic analysis were conducted.
Starks et al. (2020)	United States	Durham, North Carolina	Practical simulation	10	-	Participants performed 911 call and CPR, then attached a drone-delivered AED. Simulations were timed and video-recorded, pre- and post-simulation surveys administered.
Starks et al. (2020)	United States	Durham, North Carolina	Interviews	16	-	Participants identified based on professional position were interviewed. Qualitative data collected were analysed using NVivo, thematic and descriptive coding performed.
Zegre-Hemsey et al. (2020)	United States	17 participants from the work of Rosamond et al. (2020)	Practical simulation, interviews	17	-	Participants were paired to respond to simulated OHCA with AED drone delivery. Semi-structured qualitative interviews and audio recording analysis were conducted.
Bauer et al. (2021)	Germany	329 counties across Germany	Simulation	1427	Representative data from 31 Emergency Medical Services	Location allocation analysis used to develop three UAV networks. Cost effectiveness for each was calculated and compared to EMS.
Chu et al. (2021)	Canada	Regional Municipality of Peel in Southern Ontario	Simulation	3573	Peel Regional Paramedic Services	Mathematical optimisation model determined drone base locations from existing infrastructure. Drone response time compared to EMS response time and dispatch rules compared to ‘never dispatch’ and ‘always dispatch’ baseline policies.
Derkenne et al. (2021)	France	800 km^2^ area across Greater Paris	Simulation	3014	Sudden Death Expertise Centre Registry	Simulated time taken by basic life support team to deliver AED in OHCA events compared to time required by AED drone. OHCAs were classified into four groups and proportion of events in each group was calculated.
Ryan et al. (2021)	United Kingdom	Charlottesville-Albemarle County Area	Simulation	18	-	GIS model determined drone base placement. ArcGIS-simulated response times and distance travelled of drones compared against EMS.
Schierbeck et al. (2021)	Sweden	Controlled airspace of Save airport, Gothenburg	Prospective trial	14	-	Drones integrated in EMS for test flights, then in real-life suspected OHCAs. Proportion of successful AED drone deliveries, proportion of drone arrival before ambulance and time benefit vs. ambulance recorded.
Schierbeck et al. (2021)	Sweden	3 major counties: Stockholm, Vastra Gotaland, Skane counties	Simulation	39,246	Swedish Registry for Cardiopulmonary Resuscitation 2010 to 2018	ArcGIS spatial analyses of drone number and placement to meet coverage goals for different incidence areas performed. Simulated median timesaving of drones vs. EMS calculated per coverage goal and incidence area.
Choi et al. (2021)	South Korea	Seoul	Simulation	18,856	Korea OHCA Registry	Simulated call to AED attach times, accounting for three-dimensional topography, compared between four weather dispatch scenarios.
Rees et al. (2021)	United Kingdom	Wales	Practical simulation	6	-	Six flights and four parachute AED drops performed with an end-to-end demonstration of AED delivery via drone to simulated OHCA with bystander resuscitation.
Baumgarten et al. (2021)	Germany	Vorpommern-Greifswald rural district	Practical simulation	46	-	Participants performed CPR on a manikin, after which an AED was delivered by drone. Qualitative data from observations, interviews, and video recordings were content analysed.

GIS: geographic information system; MLCP: maximum coverage location problem; MLCP: maximum coverage location problem with complementary coverage. EMS: emergency medical services; UAV: unmanned aerial vehicle; OHCA: out-of-hospital cardiac arrest; CPR: cardiopulmonary resuscitation.

**Table 2 jcm-11-05744-t002:** Summary of Results from Simulation and Interventional Studies.

Comparison	Studies	Results
**Time saving when comparing drones and usual care**		
Drones vs. EMS	7	*Drones arrived faster than EMS in majority of OHCA cases.*
		Median reduction in response time can be as much as 16:39 min in rural areas (Claesson 2020) Time reduction of 2.06–4:24 min in rural areas (Drennan 2020) In 93% of cases, drones arrived 3:10 min faster than the EMS team (Derkenne 2020) In all test flights, drones arrived earlier than EMS, with the largest difference being 8:00 min (Cheskes, 2020) In 64% of cases, drones arrived prior to EMS with a median time difference of 1:52 min (Schierbeck 2021) Use of both drones and ambulances resulted in median time saving of 5:01 min (Schierbeck 2021)
		*The improvement in response time was greater in rural areas but lesser in urban areas.*
		Drones arrived before EMS in 32% (urban) and 93% (rural) of cases (Claesson 2020) Mean amount of time saved was 1.5 min (urban) and 19 min (rural) (Claesson 2020) Improvement in response times was more significant in rural areas, with up to 50% improvement in rural areas (Mackle 2020)
Drone vs. bystanders	1	*Drone-delivered AEDs arrived faster compared to bystander searches if AEDs were not readily accessible (Rosamond 2020).* Average time from simulated OHCA to AED delivery was 1:21 min faster by drone vs. ground search (4:45 vs. 6:06) Drone delivery was favoured when AEDs were less accessible, but ground search was favoured when AEDs were more accessible
Drone optimisation	5	*Models varying according to algorithms or optimised drone locations improved outcomes.*
		Percentage of OHCA reached <1 min was 4.3% (current EMS), 80.1% (drones launched from existing EMS stations) and 90.3% (drones launched from new sites) (Pulver 2016) To meet the 3 min response time improvement goal, the use of an integrated location-queueing model required less bases and drones compared to a region-specific model and improved median and 90th percentile time to AED (Boutilier 2019) Use of a drone dispatch rule allowed drones to reach the patient before EMS for 80.9% of cases, compared to 66.8% if drones were dispatched for all OHCAs (Chu 2020) Drone delivery of AED from a regionally optimised location was 9 km vs. 20 km (EMS) and were faster to arrive by 7–8 min compared to EMS (Drennan 2020) Machine-learning dispatch rules allowed maintained improvements, with up to 30% fewer dispatches with high accuracy (Chu 2021)
**Effects of drone models on outcomes measuring speed of response**		
Effects of varying the number of drones	1	To reach 50% of the historically reported OHCAs in <8 min, 21 drone systems would be needed; for 80%, 366; for 90%, 784, and for 100%, 2408 (Schierbeck 2020)
Effects of varying the location of drones	3	Increasing weightage of backup coverage from 0.0 to 0.2 to 1.0 required an increased number of drones (56, 71, 75) and launch sites and resulted in increased backup coverage (19%, 58.9%, 75.9%), but decreased primary coverage (Pulver 2018) 50.0% (50 stations), 83.0% (500 stations), to 96.5% (1015 stations) of OHCA response victims can be reached within 5 min (Bogle 2019) Compared to drone placement at 1st responder bases only, the use of combined placement of EMS bases and post offices improved rapid response coverage (<5 min) from 29.7% to 70.1% (Ryan 2021)
**Feasibility and cost-effectiveness of implementing drone systems**		
Cost-effectiveness	5	*Significant expenditure is needed in order to purchase and maintain drones, as well as creating suitable sites for drone bases.* SGD 50,000 required to establish a new drone launch site, SGD 10,000 required to customise an existing site, and SGD 20,000 to purchase drones (Pulver 2016) For 50 stations to reach 50% of OHCA in <5 min, the 4-year cost is SGD 1.3 million, the cost per QALY is SGD 1937; the cost per additional survivor is SGD 14,752. Achieving 96.5% requires 1015 docking stations with a 4 year cost of SGD 26.5 million, with an estimated SGD 10,438 per incremental QALY, and a cost per additional survivor of SGD 76,495 (Bogle 2019) The lifespan of a drone is 4 years (Mackle, Bauer and Bogle) and the minimum cost is typically USD 15,000 per drone The long-term maintenance cost is assumed to be 20% of the drone purchase price annually (Mackle, Bauer)
Feasibility	6	*The presence of variable terrains, technical issues, and legal restrictions impact feasibility of drone delivery.* Pulver et al. assumed that drones fly in straight lines in their simulation. Incorporation of trees and buildings could reduce service range by 10%. Choi et al. reported median flight time was 1.6 min longer in the simulator, reflecting topographical barriers as compared to the straight line distance. Success rate of call to AED attach time within 5 min of flight was reduced from 34.8% to 25.0%. Glick et al. cited technical issues, such as maintenance time, which affected drone coverage of OHCA events. Bauer, Mackle, Boutilier and Schierbeck et al. cited issues such as legal restrictions, such as airspace conflicts and no-fly zones impacting drone coverage, and permission to fly drones out-of-sight
Weather	4	*Meteorological conditions also significantly impact drone dispatch and response times.* Proper fleet sizing could address the variability in demand and weather conditions, but will not eliminate delivery failures resulting from inoperable extreme weather conditions. Ambient temperature was mostly negligible for short and strict delivery time limits (Glick 2020) Drones are unavailable for use in restrictive weather conditions (Lancaster 2020) High winds and cold temperatures also affect response times and blunt time savings (Drennan 2020) Rain and wind were the predominant prohibiting factors for flights of all planned operational hours (Schierbeck 2021) Model limiting UAV operation at night and in bad weather did not reduce call to AED attach time for any EMS station in Seoul used for UAV-AED installation (Choi 2021)

EMS: emergency medical services; OHCA: out-of-hospital cardiac arrest.

**Table 3 jcm-11-05744-t003:** Summary of Qualitative Results.

Author	Year	Key Results
Sanfridsson	2019	*Participant attitude and experience in drone delivery of AEDs* Positive setting towards using drones to deliver AED in suspected OHCA No participant hesitated or misinterpreted instructions when the dispatcher asked them to retrieve the AED from the drone No fear or hesitancy to approach drone, but sense of relief *Difficulties faced by participants, and concerns* Difficulties with mobile phone usage (calling the dispatcher, activating the speakerphone) during the simulation Difficulties in AED handling, chiefly in attachment and placement of electrodes Participant stress associated with poorer performance and compliance to dispatcher instructions Long instructive sentences by the dispatcher caused participants to stop compressions to listen to the provided information Concern from participants about finding the AED fast enough and having direct physical contact with the drone Some participants were uncomfortable with leaving the victim alone to retrieve AED; pairs felt safer and more manageable *Enabling factors in AED retrieval, attachment and usage* Dispatcher interaction provided a sense of security and support; made it easier to handle the situation and perform the given tasks Short encouraging sentences had an observed positive effect on CPR compressions Drone hovering to mark the location of AED, and the red colour of the AED bag increased ease of locating the AED Participants wished that the drone had headlights
Rosamond and Zegre-Hempsey	2020	*Participant attitude and experience in drone delivery of AEDs* A total of 89% of drone trial participants felt comfortable as the drone approached, and 72% reported no safety concerns More than half of the ground search participants reported difficulty finding the AED Generally positive feedback on drone user experiences, but neutral feelings towards interacting with the drone Overall perceived benefit of the drone delivery network in its efficiency and ability to deliver to less accessible locations Advantage of staying with victim to continue CPR without needing to search for an AED themselves *Difficulties faced by participants and concerns* Uncertainty of drone landing location Safety concerns of direct interaction with drones and landing in crowded areas *Enabling factors in AED retrieval, attachment and usage* Need for clear telecommunicator instructions
Sedig	2020	*Difficulties faced by participants, and concerns* Wariness and hesitation due to poor understanding of current paramedic services; concerned regarding possibility of drone program replacing paramedic services Lack of CPR and AED literacy Desire to be made aware of all stages of testing of the project, and for in-person demonstrations
Starks	2020	*Stakeholder attitude towards drone delivery of AEDs* Broad support for the drone network—value perceived in reduced response times and to access of hard-to-reach areas *Challenges raised by stakeholders* Operationalisation of autonomous drone AED network and financial liabilities Privacy and safety concerns; current legal and regulatory requirements Public buy-in and concern of public actually using an AED Need for research on treatment and cost-effectiveness *Facilitators of drone network development* Solidification of key partnerships, e.g., EMS and fire services Identification of viable funding Learning from existing drone models.
Baumgarten	2021	*Participant attitude and experience in drone delivery of AEDs* Bystanders and community first responders were able to collect the AED without any safety concerns A total of 8.9% of bystanders reported hesitancy to collect the AED and 2.2% found it cumbersome; none of the community first responders expressed problems A total of 95.6% of bystanders and 100% of community first responders supported the implementation of UAS-based AED delivery systems

## Data Availability

Medline: https://pubmed.ncbi.nlm.nih.gov/ (accessed on 28 February 2022). EMBASE: https://www.embase.com (accessed on 28 February 2022). Cochrane CENTRAL: https://www.cochranelibrary.com/central (accessed on 28 February 2022). PsychInfo: https://www.apa.org/pubs/databases/psycinfo (accessed on 28 February 2022). Scopus: https://www.scopus.com (accessed on 28 February 2022).

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
