# Peer review of "The Role of Drones in Out-of-Hospital Cardiac Arrest: A Scoping Review"

_jcm, 2022, doi:10.3390/jcm11195744_

Round 1

Reviewer 1 Report

Thank you very much for the opportunity  to review this nice review of the literature about the use of drone to veicolate AED to cardiac arrest victims. 

The Authors did a great job and express their findings properly. I would probably stress more the fact that the majority of the studies they  cited are simulated and some of them on the PC so from here it could difficult to take practical conclusions. Simulation is of pivotal importance but it is impossible to take really into account all the possible variable of real use. The second aspect is that without a life saving system it is difficult to save lives. I mean, an AED can be carried in different ways even in less than 1 minute but are we sure that someone who is available to use it and to perform CPR will there in the same time frame? The drone delivery system must be coupled with a good system of first responders activation in order to get on scene both the AED and someone who uses it.

Some suggestions:

1) reference one is rather dated. Please provide  more updated values of incidence

2) you might quote and discuss this papers about the use of AED across Europe showing an inhomogeneous pattern of rules and behaviors. : J Clin Med. 2021 Oct 28;10(21):5018

Author Response

Dear reviewer,

Reviewer 2 Report

With a great interest I read a review from Lim et al. on the role of drones in out-of-hospital cardiac arrest. Authors should be congratulated for a great research idea!

I would recommend following adaptations of the work:

Abstract: “Automated External Defibrillators“ are written in small capitals.

Line 24: please reformulate by removing one “and”.

Line 25: “Emergency Medical Services“ small capitals.

Line 27: Please replace number 911 with “emergency call”. Moreover, emergency number for South Korea is 112/119.

Results in abstract do not present a real presentation of overall results of the work. Please revise.

Line 30: introduce shortcut.

Introduction

Line 42: Please replace word “treatment” with resuscitation.

Line 43: Please add word may before “allow”

Line 44: Please replace word dramatically with more appropriate word.

Lines 43-45: Please reformulate this sentence, it is missing good scientific writing

Line 48: Please replace word sensibly with more appropriate word.

Line 51: Early defibrillation is one of main factors. Please revise.

Line 64: please replace word care with devices or similar appropriate word. Drones do not deliver care.

Line 65-66: Drones may be able to deliver the AEDs directly to bystanders, thus reducing the time to early defibrillation- please revise.

Line 69-70: please reformulate by removing one word “review”

Methods

Lines 78-79: please reduce repetition of words/phrases.

Why did you search “PsychInfo“ databank? This databank normaly should not contain literature on OHCA? What is the one reference that was selected? Otherwise please remove this database from your search.

Line 92: please replace word deduplications with removing duplicates

Line 93: word “view” is not appropriate here.

Line 94: instead viewed use for example assessed.

Please remove date 15.10.2021 ad hold only the date of the last search, which should be the same for all databases (28.02.2022). Reader does not profit from knowing that you performed 2 searches.

Line 99 and the whole document: more appropriate word instead of “sieve” is “asses”

Results

Line 131 and the whole text: Do not start a sentence with a number in form of a number.

Please adapt the figure 1 to fit it into main text of the manuscript. It should be presented within the manuscript and not as supplementary.

Table 1: add shortcuts at the end of the table.

Line 170: please add proper referencing.

Line 172: please provide one paragraph focusing discussion only on this.

Line 177 and further in text: please add proper referencing, avoid author names except if really needed.

Lines 198-199: please reformulate.

Perception of Drone use in AED delivery”: please avoid separation in many paragraphs. Apply this to the whole paper. The flow of text is in this form not acceptable. This should be reduced to max 2-3 paragraphs in one subsection.

Discussion

Line 267-269: is this true?

Line 271-274: please include discussion of line 172.

Line 283: please reformulate, word “vacuum” is not appropriate in English language.

Limitations need to be extended for at least 3 times as now. Please check other reviews how they did it.

Author Response

Dear reviewer,

Round 2

Reviewer 2 Report

Dear Authors,

Thank you for your reply. I have still concernes due to use of the "PsychInfo" database in the study like your. There is no "must number" of studies but recommenden. I suggest you to remove this database from your paper.